# ENUMERATE–CONJECTURE–PROVE: FORMALLY SOLVING ANSWER-CONSTRUCTION PROBLEMS IN MATH COMPETITIONS

## ABSTRACT

Mathematical reasoning is central to artificial intelligence, with applications in education, code generation, and research-level mathematical discovery. Mathematical competitions highlight two problem types: theorem-proving, requiring rigorous proofs, and answer-construction, requiring creative generation and formal verification of mathematical objects. Existing research reveals that LLMs can tackle difficult answer-construction tasks but are prone to errors from hallucinations and unverifiable steps, while symbolic methods guarantee rigor but falter in creative answer construction. This raises a key understudied question: *how to solve answer-construction problems while preserving both LLM creativity and mathematical rigor?* To address this problem, we introduce the *Enumerate–Conjecture–Prove* (ECP) framework, a modular neuro-symbolic method integrating LLM-based enumeration and pattern-driven conjecturing with formal theorem proving in Lean, and ConstructiveBench, a dataset of 3,640 formal answer-construction problems from math competitions. ECP is model-agnostic and shows consistent improvements over pure LLM baselines: on the subset of PutnamBench for answer construction, ECP formally solves 6 out of 337 answer-construction problems end-to-end (up from 4 without ECP) using GPT-5 mini and DeepSeek-Prover-V2-7B. On ConstructiveBench, ECP achieves 33.1% end-to-end state-of-the-art accuracy (up from 32.5%), demonstrating its potential to advance formal mathematical reasoning by combining LLM conjecturing with formal verification.

## 1 INTRODUCTION

Mathematical reasoning is fundamental to artificial intelligence (Newell & Simon, 1956), enabling significant progress in domains such as mathematics education, formally verified software, and mathematical research itself (Li et al., 2024). Recent advancements show that LLMs, when guided by careful prompting and tool integration, excel in solving straightforward high-school mathematics problems, but substantial challenges remain in problems requiring formal proof or the construction and verification of complex mathematical objects (Yang et al., 2024). MathArena (Balunović et al., 2025) explicitly highlights this gap, noting that state-of-the-art LLMs achieve near-perfect accuracy on answer-only contests like the American Invitational Mathematics Examination (AIME), yet consistently fail on the United States of America Mathematical Olympiad (USAMO), which demands rigorous proofs and creative answer construction.

Mathematical competitions, in particular, present two challenging problem types: theorem-proving, requiring rigorous proofs of stated conclusions, and answer-construction, involving hypothesizing and formally verifying mathematical objects. In theorem-proving problems, a clear conclusion is stated, and the goal is a formal proof based on provided hypotheses. Conversely, answer-construction problems require the construction of specific mathematical objects (e.g., functions, numbers, sets) that satisfy given conditions, along with proofs of correctness. Representative examples of these problems appear in Table 1. Crucially, answer-construction problems present the unique challenge of identifying the correct candidate object or structure prior to formal verification.

We distinguish between *informal mathematics*, characterized by intuitive reasoning and natural language arguments, and *formal mathematics*, represented explicitly in machine-verifiable proof

*Answer-Construction Problem:*
Find all functions $f : \mathbb{R}^+ \to \mathbb{R}^+$ such that $\forall x, y \in \mathbb{R}^+$,

$$x\big(f(x) + f(y)\big) \geq \big(f(f(x)) + y\big)f(y)$$

*Answer:* All $f(x) = \frac{c}{x}$ for some $c > 0$.
*Proof:* ...

*Theorem-Proving Problem:*
Let $x_1, \ldots, x_{2023}$ be distinct positive numbers such that for all $n \in \{1, \ldots, 2023\}$,

$$\sqrt{\left(\frac{1}{x_1} + \cdots + \frac{1}{x_n}\right)(x_1 + \cdots + x_n)} \in \mathbb{Z}$$

Prove that $x_{2023} \geq 2024$.
*Proof:* ...

Table 1: Examples of competition-level problems. The left panel shows an answer-construction problem requiring a candidate answer and proof. The right panel shows a standard theorem-proving problem.

assistants such as Lean (Moura & Ullrich, 2021). Autoformalization is an automated process that translates informal problem statements into formal representations. While LLMs are effective at creative and exploratory reasoning, they frequently generate unverifiable and incorrect assertions due to hallucinations (Huang et al., 2025). Symbolic reasoning tools, such as SMT solvers (e.g., Z3 (De Moura & Bjørner, 2008)), serve as the logical backbone of automated reasoning tools but cannot feasibly discover closed-form answers in the vast space of candidates.

Inspired by Pólya's problem-solving methodology (Polya, 2014), which emphasizes systematically exploring simpler cases, identifying patterns, and rigorously proving general cases, we introduce the ECP framework. Our framework combines the exploratory strengths of LLMs, which enumerate candidate answers through programmatic execution and generalize patterns to form conjectures, with the rigor of formal theorem-proving methods that verify these conjectures. ECP thus effectively addresses the critical limitation of prior approaches by formally solving answer-construction problems, filling a significant gap in the mathematical reasoning landscape.

Our contributions are twofold: (i) ECP, a modular framework that unifies LLM-driven code generation for enumeration, pattern-based conjecture for answer construction, and formal theorem proving in Lean; (ii) ConstructiveBench, an autoformalized dataset of 3640 competition-level answer-construction problems with verified Lean formalizations, validated through experiments with LLMs and state-of-the-art theorem-proving techniques.

## 2 RELATED WORK

**Formal Theorem Proving**  Formal methods such as SMT solvers (e.g., Z3 (De Moura & Bjørner, 2008), CVC5 (Barbosa et al., 2022)) and first-order provers (e.g., Vampire (Kovács & Voronkov, 2013), E (Schulz, 2002)) are widely used for formal verification and logical reasoning. Interactive theorem provers (ITPs) like Isabelle (Paulson, 1994) and Lean (Moura & Ullrich, 2021) allow step-by-step formal proof development, often integrating automation via tools such as Sledgehammer (Böhme & Nipkow, 2010) and LeanCopilot (Song et al., 2024).

Recent neuro-symbolic systems integrate LLMs with formal proof environments. Goedel-Prover (Lin et al., 2025), the DeepSeek-Prover series (Xin et al., 2024; Ren et al., 2025), InternLM-Math (Ying et al., 2024b), SeedProver (Chen et al., 2025), Kimina-Prover (Wang et al., 2025) and AlphaProof (Teams, 2024) scale formal data generation and RL-style training to improve Lean proof success on miniF2F, PutnamBench, and related evaluations. Domain-specific systems, including AlphaGeometry (Trinh et al., 2024; Chervonyi et al., 2025) and PyEuclid (Li et al., 2025b) for Euclidean geometry, LIPS for inequalities (Li et al., 2025a), target particular subfields with symbolic search and automation. Beyond theorem proving, Liu et al. (2025b) introduce a search-based formal problem-solving framework (FPS/D-FPS) with new benchmarks for process-verified solutions.

**LLMs for Mathematical Reasoning**  Beyond formal proof generation, the broader math-reasoning literature explores prompting, learning, and tool use. Prompting-based approaches (e.g., chain-of-thought) encourage explicit intermediate reasoning (Wei et al., 2023). Reinforcement learning has recently become central: GRPO—popularized in Shao et al. (2024)—and large-scale systems like DAPO (Yu et al., 2025a) improve step-wise reasoning via preference/advantage signals at scale. Com-

plementary to policy learning, tool-integrated multi-turn training frameworks such as ReTool (Feng et al., 2025) explicitly train agents to plan, call tools, and verify intermediate computations. Orthogonal "code-as-tool" frameworks, including PAL (Gao et al., 2023) and Tora (Gou et al., 2023), and logic-solver integration such as SATLM (Ye et al., 2023), demonstrate the benefits of executing verifiable intermediate steps. Liu et al. (2025b) propose a broader formal problem-solving framework (FPS/D-FPS) with new benchmarks for process-verified solutions.

**Evaluation Benchmarks** We group evaluation datasets into informal and formal categories. Early works in mathematical evaluations, such as GSM8K (Cobbe et al., 2021), MATH (Hendrycks et al., 2021), and MathQA (Amini et al., 2019) assess numerical/short-answer problem solving in natural language. Follow-up works include harder answer-only suites like OlympiadBench (He et al., 2024), Omni-Math (Gao et al., 2024a), and MathOdyssey (Fang et al., 2024) though they still rely on string matching for answer verification. In parallel, formal benchmarks like miniF2F (Zheng et al., 2022), FIMO (Liu et al., 2023), PutnamBench (Tsoukalas et al., 2024), ProofNet (Azerbayev et al., 2023), LeanWorkBook (Ying et al., 2024a), Herald (Gao et al., 2024b), and the recent FormalMATH (Yu et al., 2025b) enable machine-verifiable evaluation in Lean/HOL. ProverBench (Xin et al., 2024) contributes a mixed set of 325 formalized problems, including AIME-style items, expanding coverage toward competition math. MathConstruct (Balunović et al., 2025) targets constructive proof tasks. CombiBench (Liu et al., 2025a) focuses on combinatorics in Lean.

## 3 METHODS

### 3.1 PROBLEM FORMULATION

Problems in math competitions can generally be classified into two categories: *theorem-proving tasks* and *answer-construction tasks*. In a theorem-proving task, the problem explicitly presents a set of hypotheses $P$ and a target conclusion $Q$, and the objective is to formally prove that $P$ implies $Q$. Prior benchmarks and research on AI for formal mathematics, such as MiniF2F (Zheng et al., 2022) and FIMO (Liu et al., 2023), focus on this theorem-proving task format, which aligns with traditional first-order theorem proving.

In contrast to theorem-proving tasks, an *answer-construction* task asks for the construction of mathematical objects satisfying specified constraints, without an explicitly stated conclusion. We define an answer-construction task by the following components in the context of Lean dependent type theory:

**Context Variables:** The unassigned context variables $a := (a_1, \ldots, a_k)$ (with type $\alpha := \alpha_1 \times \cdots \times \alpha_k$) in the problem.

**Context Property:** The predicate $P : \alpha \to \text{Prop}$ that the context variables satisfy.

**Answer Variables:** The unknown answer $b$ (with type $\beta$) to be constructed.

**Answer Function:** $f : \alpha \to \text{Set } \beta$ mapping each context instance to a set of valid answers.

**Problem Predicate:** A Boolean predicate $Q : \alpha \times \beta \to \text{Prop}$ such that $P(a, b) = \text{True}$ if and only if $b$ is a valid answer under context $a$.

We demonstrate the definition with a simple, straightforward example: "Given $a, b \in \mathbb{Z}$ with $b \neq 0$, find all $x \in \mathbb{R}$ such that $ax = b$." Here, the context variables are $(a, b)$ with type $\mathbb{Z} \times \mathbb{Z}$ and property $P(a, b) := a \neq 0$. The answer variable is $x$ with type $\mathbb{R}$, the problem predicate is $Q(a, b, x) := ax = b$, and the answer function is $f(a, b) = \{\frac{b}{a}\}$.

In general, the task of answer-construction can be formulated in Lean's dependent type theory as follows:

$$\exists f : \alpha \to \text{Set } \beta, \ \forall(a : \alpha), \ P(a) \to \forall(b : \beta), \ Q(a, b) \leftrightarrow b \in f(a).$$

Intuitively, the task is to construct a function $f$ such that for any context parameter $a$ that satisfies property $P$, $f(a)$ precisely captures all and only the valid answers. Variants of format constraints on answers can be incorporated by refining the problem predicate $Q$. For example, to enforce minimality, it is sufficient to define $Q_{min}(a, b) := P(a, b) \land \forall c < b, \ \neg P(a, c)$ or use Lean's built-in notions

such as IsLeast or IsGreatest. When the problem admits a single unique solution, $f$ degenerates to $\alpha \rightarrow \beta$, and it becomes sufficient to show

$$\exists f : \alpha \rightarrow \beta, \ \forall(a : \alpha), \ P(a) \rightarrow \big(\forall(b : \beta), \ Q(a,b) \leftrightarrow b = f(a)\big).$$

When there are no context variables, the formulation reduces to $\exists f : \text{Set } \beta, \ \forall b : \beta, \ P(b) \leftrightarrow b \in f$.

To ensure answers are not only correct but also semantically meaningful, we further require that constructed answers be expressed in closed-form canonical format without echoing the problem statement's syntax or structure. For example, in the following task: `theorem test: (x y : ℕ) (hpos : 0 < x ∧ 0 < y) : x ^ 3 + y ^ 3 = x ^ 2 + 42 * x * y + y ^ 2 ↔ (x, y) ∈ answer` the answer `{(7, 1), (1, 7), (22, 22)}` is legal, but `{(x,y):ℕ | x ^ 3 + y ^ 3 = x ^ 2 + 42 * x * y + y ^ 2}` is illegal because it trivializes the problem.

The formal answer-construction task is expressed as a theorem statement with a placeholder or an existentially quantified answer. The autoformalization process will be discussed in Section 4.3 in detail.

## 3.2 ENUMERATE–CONJECTURE–PROVE FRAMEWORK

ECP enables LLM to perform multi-turn tool calls with Python and Lean in the ReAct manner (Yao et al., 2023). The framework and structured output are implemented using the APPL prompt programming language (Dong et al., 2024), and we include the prompt template in Appendix B. Figure 1 visualizes the workflow of ECP applied to an example answer-construction problem end-to-end.

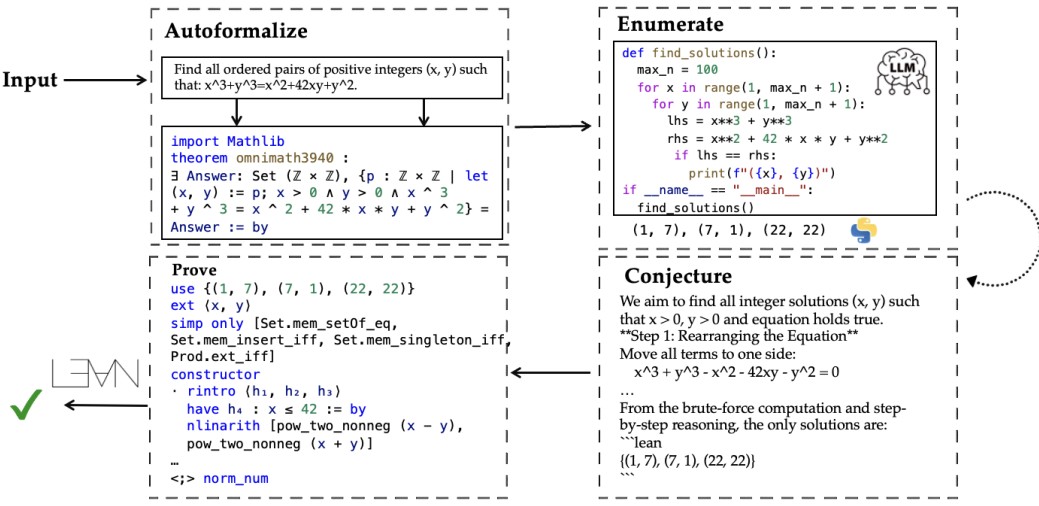

Figure 1: Illustration of the ECP framework applied to a formalized Balkan MO Shortlist problem in OmniMath dataset (Gao et al., 2024a).

**Enumerate Stage.** In the initial Enumerate stage, the LLM is given a formal answer-construction task, and the goal is to enumerate the candidate answers by generating Python programs. The model performs multi-turn interaction to refine the program and produce enumeration: {<reasoning_1> <python_call_1> <python_response_1><reasoning_2>...}. For problems with unassigned context variables or infinitely many answers, the model is encouraged to explore small customized setups and return bounded enumerations, guided by the system prompt with constraints of timeout=60s and a maximum of 100 enumerated answers.

**Conjecture Stage.** In the Conjecture (i.e. answer-construction) stage, given the answer-construction task and the enumerations as hint, the LLM reasons about the problem, generalizes beyond enumeration and attempts to propose a syntactically valid closed-form answer in Lean, following the workflow {<reasoning_1> <lean_call_1> <lean_response_1><reasoning_2> ...}. To prevent potential answer "hacking" described in Section 3.1, we design the system prompt to explicitly disallow such illegal answers that lead to trivial or vacuous versions of the Lean theorem statement.

**Prove Stage.** In the final Prove stage, the candidate answer is substituted into the theorem statement, reducing the answer-construction problem into a standard theorem-proving task. Our framework supports invoking theorem-proving models, such as DeepSeek-Prover-V2 (Ren et al., 2025), Kimina-Prover (Wang et al., 2025), and Goedel-Prover (Lin et al., 2025), to generate proofs of the theorem, and Lean verifies whether the proof closes the goal. Additionally, we implement pure symbolic methods as a sequence of automation tactics as follows, which will be used to verify equivalence between constructed and ground-truth answer in evaluation:

1. `simp`: Simplifies expressions by rewriting.

2. `aesop` (Limperg & From, 2023): Proves the goal via a best-first search proof engine.

3. `nlinarith`: Solves nonlinear arithmetic problems over the integers and rationals.

4. `ring`: Proves equalities in commutative (semi)-rings.

5. `norm_num`: Evaluates and proves numeral equalities and inequalities.

Successful Lean verification of the proof concludes the ECP pipeline.

## 4 CONSTRUCTIVEBENCH DATASET

In this section, we introduce the ConstructiveBench dataset, detailing its sources, curation methodology, and contents, alongside special considerations such as data contamination and a human evaluation of autoformalization quality.

### 4.1 DATA SOURCES AND SCOPE

ConstructiveBench is curated by aggregating and rigorously filtering problems from reputable mathematical sources. Primary sources include official competition archives such as AMC 12 A/B (from 2000 onward), AIME I/II, HMMT, regional Olympiads, and the IMO Shortlist and Longlist. Additionally, we integrate data from established informal math datasets including *OlympiadBench* (He et al., 2024), *Omni-Math* (Gao et al., 2024a), and *MathOdyssey* (Fang et al., 2024). We employ structural filtering to retain exclusively *answer-construction* problems—those requiring numerical or symbolic answers alongside proofs—and exclude purely theorem-proving tasks without explicit answer-construction components.

### 4.2 DATA DEDUPLICATION

The raw source dataset contains 5,555 problems. To remove duplicates, we apply both string-based matching and embedding-based filtering using Sentence-BERT (Reimers & Gurevych, 2019) and FAISS (Douze et al., 2024) with a 90% semantic-similarity threshold. This process yields 5,316 unique problems spanning a broad difficulty range—from challenging high-school problems to sophisticated Olympiad-style questions that cover diverse domains including algebra, number theory, combinatorics, probability, and calculus.

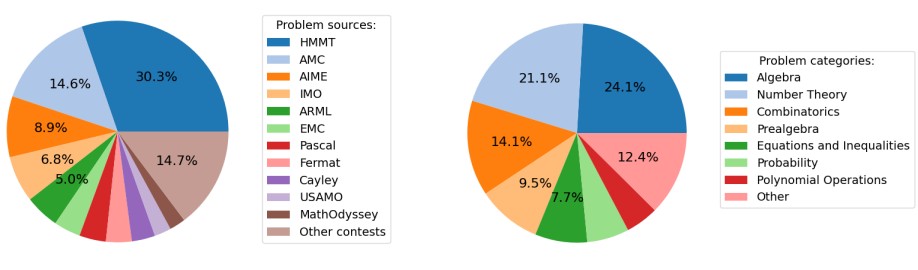

(a) **Problem sources**          (b) **Problem domains**

Figure 2: Problem sources and domains in ConstructiveBench. Only the top 11 sources and top 7 domains are shown; the remainder are grouped under 'Other'.

### 4.3 PROBLEM AUTOFORMALIZATION

To formally solve answer-construction problems in Lean, we first autoformalize each informal problem into the formal task described in Section 3.1.

**Autoformalization with LLMs.** We leverage state-of-the-art autoformalization models *Herald-Translator* (Gao et al., 2024b) and *Kimina-Autoformalizer-7B* (Wang et al., 2025), together with the commercial model GPT-5 mini using a 3-shot prompt. We also use GPT-5 mini as an LLM judge. A candidate formalization is accepted if and only if both conditions hold: (i) the Lean compiler returns no errors; and (ii) the LLM judge deems the formalization semantically equivalent to the informal statement, with correct use of key definitions and no trivial restatement.

**Information Retrieval in Lean.** To mitigate hallucinations and errors arising from scarce aligned informal-to-formal data (Wu et al., 2022) and rapidly evolving Lean syntax, we construct an external Lean knowledge base and an interactive loop for interaction between LLM and Lean. The knowledge base contains approximately 385K entries from the Lean 4.23.0 documentation and the `mathlib` repository (mathlib Community, 2020). After filtering to retain entries most relevant to high-school-level formalizations (Appendix A), the database is reduced to 57K items. Each entry is embedded with Sentence-BERT (Reimers & Gurevych, 2019) and indexed by FAISS (Douze et al., 2024); when error messages indicate missing or hallucinated definitions, we retrieve the top 5 candidates by semantic similarity and the top 5 by edit distance on the symbol name.

**Error-Guided Feedback Loop.** Each of the three LLMs receives the informal problem with a 3-shot prompt and drafts a formalization. Lean compiler error messages, retrieved definitions, and LLM-judge feedback are then fed back to the model for iterative refinement, up to $T = 5$ iterations. A problem is considered successfully autoformalized if *any* model produces an accepted formalization within the budget (a best-of-$N$ strategy across models).

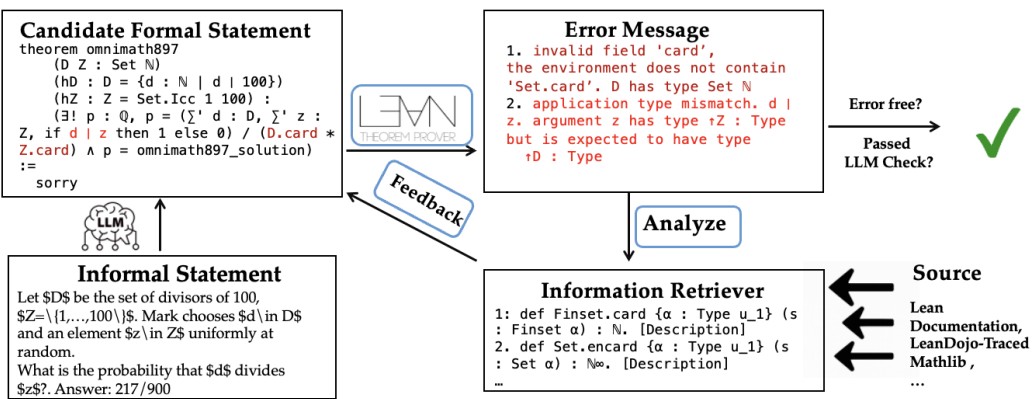

Figure 3: Overview of the autoformalization pipeline: the LLM drafts a formal statement; when compilation or semantic checks fail, error messages and retrieved Lean references are fed back; the loop repeats until both checks pass.

### 4.4 DATASET VALIDATION

**Outputs and Coverage.** Starting from 5,316 deduplicated informal problems, we run up to $T = 5$ refinement iterations per model. This produced 4,729 Lean-compilable formalizations in the correct format; among these, 3,640 also pass the LLM-judge semantic check and constituted the final ConstructiveBench dataset. Each accepted entry includes the informal statement, the formal Lean theorem, the ground-truth answer, an informal solution, and metadata such as source and difficulty. Figure 4 shows an example dataset entry and the distribution of answer types. The diversity of answer types enables comprehensive evaluation across a wide range of problem formats.

**After-Cutoff Subset.** To mitigate contamination from pretraining corpora—where LLMs might inadvertently memorize training examples—we include a test split of 106 problems created after

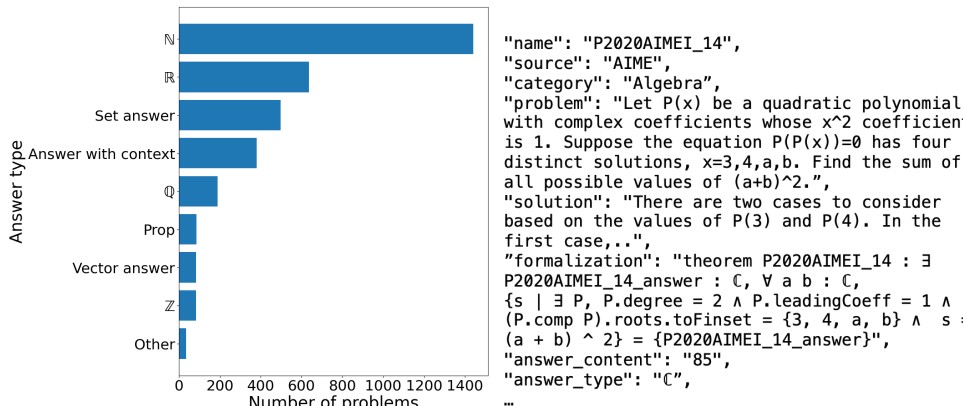

Figure 4: **Left**: Distribution of answer types in ConstructiveBench (top 8 shown; the remainder are grouped as 'Other'). **Right**: Example dataset entry.

June 2024 (beyond the knowledge cutoff date of most LLMs we will evaluate), ensuring fair evaluation on unseen problems.

**Human Evaluation.** To estimate the quality of autoformalized theorem statements in ConstructiveBench, we conducted a human evaluation on a random slice of 100 formalizations from the 3,640 accepted problems. Following the rubric in Gao et al. (2024b), each formalization was labeled as *completely correct*, *minor error*, or *major error*. We found 77 were fully correct, 6 had minor errors, and 17 had major errors; examples and analyses appear in Appendix C. This result aligns with human evaluations of existing autoformalized benchmarks (66.9% accuracy on Herald (Gao et al., 2024b) and 72.1% on FormalMath (Yu et al., 2025b)), highlighting the high quality of ConstructiveBench.

## 5  EMPIRICAL EVALUATION

We evaluate whether ECP outperforms a classic chain-of-thought (CoT) baseline on two tasks:

1. **Answer-Construction.** Given a formal answer-construction problem, can the conjecturer produce a Lean expression provably equivalent to the ground truth under a fixed set of automation tactics?

2. **End-to-End Problem Solving.** Given a formal answer-construction problem, can the system both conjecture a correct answer and close the Lean goal using a prover?

### 5.1  EXPERIMENTAL SETUP

**Datasets.** We evaluate on ConstructiveBench (3,640 Lean-verified answer-construction problems) and the PutnamBench answer-construction subset (337 problems). We additionally report a contamination-controlled *After-Cutoff* split of 106 ConstructiveBench problems (created after June 2024).

**Models and Hyperparameters.** For conjecturing we use GPT-5 (mini, nano), GPT-4.1 (mini, nano), and DeepSeek (V3.1, V3). For reasoning models (GPT-5 families and DeepSeek-V3.1) and lightweight non-reasoning models (GPT-4.1 families and DeepSeek-V3), we set reasoning_effort to medium and None, respectively. We use each model's recommended decoding default parameters; for sampling we set max_tokens=4096, coder_max_attempt=3, and conjecturing_attempt=5. For proving, we use DeepSeek-Prover-V2-7B, Goedel-Prover-SFT, and Kimina-Prover-Preview-7B, with hyperparameters top_p=0.95, temperature=1.0, and max_tokens=4096. We evaluate using Pass@32 with a 120s Lean verification timeout.

**Metrics.** Answer-construction accuracy marks a problem correct if the conjectured answer $a$ is provably equal to the ground truth $a_{\text{gt}}$ via the following Lean automation tactics: simp, aesop (Limperg & From, 2023), nlinarith, ring, and norm_num. End-to-end accuracy marks a problem correct if the answer is constructed by the conjecturer and the prover produces a Lean-verified proof to the theorem statement after substituting the conjectured answer. We use Pass@32 for prover, meaning that each prover model is sampled 32 outputs independently and is marked correct if at least one proof passes Lean check. Given the fixed model choice, union accuracy simply measures the fraction of successfully solved problems by any of CoT and ECP.

## 5.2 ANSWER-CONSTRUCTION ACCURACY

In this section, we analyze the results of formal answer-construction accuracies with different conjecturer models on ConstructiveBench and PutnamBench shown in Table 2 and Table 3. Complete results for six models, including evaluations on the after-knowledge-cutoff dataset, are listed in Appendix D.

Results show that ECP consistently improves answer construction across conjecturers and datasets. On ConstructiveBench, GPT-5 mini rises from 69.7% to 73.6%, and DeepSeek-V3.1 increases from 65.4% to 75.0%, indicating that stronger conjecturers still benefit from enumeration signals. The largest absolute gains occur for lightweight non-reasoning models: DeepSeek-V3 increases from 17.3% to 40.7%. These patterns persist on the after-knowledge-cutoff split; for example, DeepSeek-V3 improves from 44.3% to 67.9%, showing robustness of ECP when data contamination is controlled.

On PutnamBench, absolute accuracies are lower due to the difficulty of the dataset, but ECP remains beneficial for all models. DeepSeek-V3.1 achieves state-of-the-art performance with 210 correct answer constructions via ECP, up from 181 problems with the CoT baseline.

In addition, analysis of union accuracy shows that CoT and ECP correctly construct answers for overlapping but distinct subsets. For example, with DeepSeek-V3.1 as conjecturer, ECP can construct correct answers for 13.8% problems that CoT baseline fails. Still, 4.2% problems fail by ECP but successful with CoT baseline because ECP's enumeration may under-generalize or fail on several problems, misleading the conjecturer in answer construction.

| Conjecturer | Answer Construction | | End-to-End Problem Solving | |
| --- | --- | --- | --- | --- |
| | CoT / ECP | Union | CoT / ECP | Union |
| GPT-5 mini | 69.7% / **73.6%** | 78.9% | 32.5% / **33.1%** | 35.0% |
| DeepSeek-V3 | 17.3% / **40.7%** | 43.4% | 10.1% / **18.0%** | 18.7% |
| DeepSeek-V3.1 | 65.4% / **75.0%** | 79.2% | 31.1% / **32.7%** | 34.3% |

Table 2: Answer-construction accuracy and end-to-end problem solving accuracy of CoT baseline and ECP on ConstructiveBench (3640 problems in total). The first column shows the conjecturer models for answer construction, and the second column shows the answer-construction accuracies under CoT and ECP methods, along with their union accuracy. the third column shows end-to-end accuracies using DeepSeek-Prover-V2-7B for proof generation.

| Conjecturer | Answer-Construction | | End-to-End Problem Solving | |
| --- | --- | --- | --- | --- |
| | CoT / ECP | Union | CoT / ECP | Union |
| GPT-5 mini | 185 / 194 | 242 | 4 / **6** | 6 |
| DeepSeek-V3 | 64 / 107 | 124 | 1 / **3** | 3 |
| DeepSeek-V3.1 | 181 / 210 | 248 | 4 / 4 | 5 |

Table 3: Number of correct answer constructions and end-to-end solved problems on PutnamBench (337 problems in total). The first column shows the conjecturer models for answer construction, with CoT and ECP results as well as their union. The second column shows the number of problems solved end-to-end with the DeepSeek-Prover-V2-7B prover (Pass@32).

## 5.3 END-TO-END PROBLEM-SOLVING ACCURACY

In this section, we discuss the results of end-to-end problem solving across various combinations of conjecturer and prover models. The rightmost columns of Table 2 and Table 3 report accuracies using different conjecturer models for answer construction, combined with DeepSeek-Prover-V2-7B for proving. The complete full per-prover tables (including Goedel-Prover-SFT and Kimina-Prover-Preview-Distill-7B) for each dataset are placed in Appendix D.

Overall, DeepSeek-Prover-V2-7B outperforms the other two prover models in these benchmarks. The strongest end-to-end problem-solving result on ConstructiveBench pairs GPT-5 mini as the conjecturer with DeepSeek-Prover-V2-7B as the prover, reaching 33.1% (up from 32.5% with CoT), with DeepSeek-V3.1 a close second at 32.7% (up from 31.1%). Even on the difficult PutnamBench answer-construction problems, ECP still solves 6 out of 337 problems end-to-end with GPT-5 mini and DeepSeek-Prover-V2-7B, compared to 4 out of 337 for CoT.

The lower overall accuracy and smaller improvements are mainly due to two factors. First, incorrectly constructed answers lead to logically wrong statements that waste prover effort. Second, the prover models have limited power and often lack the capability to formally prove difficult theorems, especially ones in PutnamBench. Overall, ECP still provides consistent end-to-end gains over CoT, with the best-performing configuration given by GPT-5 mini as conjecturer and DeepSeek-Prover-V2-7B as prover.

## 6 LIMITATIONS AND FUTURE WORK

**Quality of Autoformalized Theorem Statements.** The ConstructiveBench depends on the quality of black-box LLM autoformalizations and LLM judge. In particular, quantitatively evaluating the quality of autoformalized statements remains a challenge in AI for mathematics. While manual evaluation offers high fidelity and we evaluated the autoformalization quality of a subset of 100 problems, it is still time-consuming and intractable to large scale datasets. Proxy metrics such as compile success rates or BLEU scores are more efficient but often fail to capture logical equivalence with the original problem. Future work should explore improved evaluation metrics for formalization quality and expand datasets that better align informal and formal representations of mathematical problems.

**Theorem-Proving.** State-of-the-art LLM is capable of winning gold medal in IMO exam (Huang & Yang, 2025), yet this achievement is out of reach for symbolic and LLM-based theorem provers in formal environment such as Lean. Our proving stage leverages off-the-shelf theorem provers, which still leaves significant room for optimization. Designing specialized domain-specific languages with inference rules tailored to high-school Olympiad mathematics could improve both success rates and runtime. Another limitation is the combinatorial explosion during the enumeration phase for difficult IMO problems, where the time complexity of the enumeration program grows rapidly as the in-context parameter increases. Scaling beyond enumeration demands insight and creativity akin to expert mathematicians, guiding the search toward promising conjectures rather than exhaustively exploring the entire solution space.

## 7 CONCLUSION

We introduced the Enumerate–Conjecture–Prove (ECP) framework, a modular neuro-symbolic approach that combines LLM-driven code enumeration, answer construction, and formal theorem proving in Lean. To support this work, we curated ConstructiveBench, a dataset of 3,640 answer-construction problems formally verified through an iterative autoformalization pipeline with error feedback. In empirical evaluations, ECP improved the state of the art on ConstructiveBench, raising GPT-5 mini's answer-construction accuracy from 69.7% to 73.6% and its end-to-end problem-solving accuracy from 32.5% to 33.1%. On the PutnamBench subset, ECP solves 6 out of 337 problems compared to 4 for the CoT baseline. These results demonstrate that systematically integrating enumeration and conjecture with formal verification can significantly advance the state of the art in competition-level mathematical reasoning. Future research includes enhancing autoformalization alignment and developing specialized symbolic provers tailored to Olympiad and research-level mathematical problems.

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

## USE OF LARGE LANGUAGE MODELS

We used LLMs to refine grammar in the paper and to assist with code implementation. All outputs were reviewed and verified by the authors.

## REPRODUCIBILITY STATEMENT

We uploaded an anonymous repository for dataset and code implementation to the supplementary materials, with a detailed README describing how to run the experiments.

## APPENDIX

## A   LEAN DEFINITION FILTERING

To filter out the rarely-used definitions in autoformalization, we prioritize the definitions and theorems within the following namespaces in Lean:

```
Nat, Int, Rat, Real, Complex, ENat, NNReal, EReal, Monoid, CommMonoid,
Group, CommGroup, Ring, CommRing, Field, Algebra, Module, Set,  Finset,
Fintype, Multiset, List, Fin, BigOperators, Filter, Polynomial, Order,
SimpleGraph, Equiv, Embedding, Injective, Surjective, Bijective, Topology
```

# B    PROMPT

## B.1    PROMPT FOR AUTOFORMALIZATION

> **LLM Prompt**
>
> You are given a math problem that requires answer-construction.
> Your task is to formalize the problem in Lean 4 (v4.23.0).
>
> 1. Start with imports and namespaces such as `import Mathlib` and `open Real`.
>
> 2. Define the ground-truth answer as an abbrev:
>    `abbrev <problem_name>_answer : $\beta$ := <expression>`
>    or
>    `abbrev <problem_name>_answer : $\alpha \to \beta$ := fun x => <expression>`
>    (Don't put `(n : $\mathbb{N}$)` before `: $\alpha \to \beta$`; encode inputs inside `fun x =>
>     ...`.)
>
> 3. Theorem skeleton (leave proof as sorry):
>    `theorem <problem_name> (x : $\alpha$) (hypotheses about x) (y : $\beta$) :
>        P(x, y) $\leftrightarrow$ y = <problem_name>_answer x := by sorry`
>
> Other Suggestions:
> - Use `IsLeast` / `IsGreatest` for optimal answers.
> - Use `Set $\beta$` and membership if there are multiple answers.

## B.2    PROMPT FOR ENUMERATOR

> **LLM Prompt**
>
> You will be given a math problem that requires answer construction in Lean.
> Your task is to write a Python program to enumerate possible answers to
> assist in conjecturing and proving the true answer.
> For problem with unfixed context parameters or infinite answers, try
> several parameters and bound the number of enumerations to 100.

## B.3    PROMPT FOR CONJECTURING

> **LLM Prompt**
>
> You will be given a math problem that requires answer construction in Lean.
> Your task is to reason about the problem and clearly state the
> closed-form final answer in Lean exprssion with explanations.
> You do not have to derive the whole proof. The program enumeration can be
> used as a hint.
>  You must not "cheat" by an answer echoing problem statement.

## C  HUMAN EVALUATION EXAMPLES

To verify dataset quality and estimate the false-positive error rate, we randomly sampled 100 formalizations (from the 3,640 problems which have passed both the compiler and LLM judge) for manual evaluation, labeling each as: - **Correct formalization** - **Formalization with minor error** - **Formalization with major error**

Below, we illustrate one example for each category. For clarity, we separate the answer and the statement, and use a placeholder for the answer variable in the Lean formalization.

**1. Correct formalization:** `omnimath19` **(Chinese Mathematical Olympiad).**  *Problem.* Let $f : X \to X$, where $X = \{1, 2, \ldots, 100\}$, be a function satisfying: 1. $f(x) \neq x$ for all $x = 1, 2, \ldots, 100$; 2. For any subset $A \subseteq X$ such that $|A| = 40$, we have $A \cap f(A) \neq \emptyset$. Find the minimum $k$ such that for any such function $f$, there exists a subset $B \subseteq X$ with $|B| = k$ and $B \cup f(B) = X$. The answer is 69.

*Formalization.*

Listing 1: omnimath19 (correct formalization)

```
import Mathlib
open Set Function
abbrev omnimath19_answer : ℕ := 69

theorem omnimath19 :
  IsLeast {k : ℕ | 0 < k ∧
    ∀ (f : Fin 100 → Fin 100),
      (∀ x, f x ≠ x) →
      (∀ A : Finset (Fin 100), A.card = 40 → (A ∩ A.image f).Nonempty) →
      ∃ B : Finset (Fin 100), B.card = k ∧ B ∪ B.image f = Finset.univ }
  omnimath19_answer := by
  sorry
```

*Analysis.* The constraint $X = \{1, \ldots, 100\}$ is equivalently captured by `Fin 100`. The condition 1, $f(x) \neq x$, is formalized in (∀ x, f x ≠ x). The constraint 2, $|A| = 40$ and $A \cap f(A) \neq \emptyset$ is captured in using `Finset.image`, `Finset.card` and `Nonempty` syntax. The conditions regarding set $B$ is correctly formalized in similar manner. Finally, the target minimality condition is encoded using `IsLeast`. Overall, the formalization is correct without error.

**2. Formalization with minor error:** `omnimath1410` **(HMMT).**  *Problem.* For positive integers $a, b$, let $M(a, b) = \frac{\text{lcm}(a,b)}{\gcd(a,b)}$, and for $n \geq 2$ define

$$x_n = M(1, M(2, M(3, \ldots, M(n-2, M(n-1, n)) \ldots))).$$

Compute the number of positive integers $n$ such that $2 \leq n \leq 2021$ and $5x_n^2 + 5x_{n+1}^2 = 26x_n x_{n+1}$. The answer is 20.

Listing 2: omnimath1410 (formalization with minor error)

```
import Mathlib
open Nat

abbrev omnimath1410_answer : ℕ := 20

def M (a b : ℕ) : ℕ := Nat.lcm a b / Nat.gcd a b

def x : ℕ → ℕ
| 0 => 0
| 1 => 1
| n + 2 => M (n + 1) (x (n + 1))   -- off-by-one here on purpose

theorem omnimath1410 :
  Finset.card
    (Finset.filter
```

```
        (fun n => 5 * (x n)^2 + 5 * (x (n + 1))^2 = 26 * x n * x (n + 1))
        (Finset.Icc 2 2021))
  = omnimath1410_answer := by
  sorry
```

*Analysis.* The definition of $M$ and the theorem statement are correctly formalized, but the recursive definition of $x$ is off by one. Specifically, in this formalization $x_2 = M(1, x(1)) = 1$, but the intended definition gives $x_2 = M(1, 2) = 2$. This can be fixed by changing `M (n + 1) (x (n + 1))` to `M (n + 2) (x (n + 1))`.

**3. Formalization with major error:** IMO2023SLN7 **(IMO).** *Problem.* Let $a, b, c, d$ be positive integers satisfying

$$\frac{ab}{a+b} + \frac{cd}{c+d} = \frac{(a+b)(c+d)}{a+b+c+d}.$$

Determine all possible values of $a + b + c + d$. The answer is *all positive integers that are not square-free.*

Listing 3: P2013AIMEII6 (formalization with major error)

```
import Mathlib
open Nat
noncomputable abbrev IMO2023SLN7_answer : Set ℕ := { n | ∃ p : ℕ, 2 ≤ p ∧ p * p |
    n }
theorem IMO2023SLN7(a b c d : ℕ)(ha : 0 < a) (hb : 0 < b) (hc : 0 < c) (hd : 0 < d)
    :( (↑a * ↑b) / (↑a + ↑b) + (↑c * ↑d) / (↑c + ↑d) = ( (↑a + ↑b) * (↑c + ↑d) ) /
    (↑a + ↑b + ↑c + ↑d) ) ↔(a + b + c + d) ∈ IMO2023SLN7_answer := by sorry
```

*Analysis.* The logical shape is completely wrong. The problem asks for all possible values of $a + b + c + d$ that occur for some positive integers satisfying the equation. However, the formalized theorem states that for some specific $a, b, c, d > 0$ the equation holds iff $a + b + c + d \in answer$, which would mean the equation is true for those exact $a, b, c, d$ precisely when the sum is not square-free.

## D FULL TABLES

### D.1 CONSTRUCTIVEBENCH FULL (3640 PROBLEMS)

| Conjecturer | DeepSeek-Prover-V2-7B | | | Goedel-Prover-SFT | | | Kimina-Prover-Preview-7B | | |
|---|---|---|---|---|---|---|---|---|---|
| | CoT | ECP | Union | CoT | ECP | Union | CoT | ECP | Union |
| GPT-5 nano | 28.8% | **30.9%** | 34.5% | 24.9% | **26.1%** | 29.5% | 22.3% | **23.2%** | 27.4% |
| GPT-5 mini | 32.5% | **33.1%** | 35.0% | 27.2% | **27.6%** | 29.1% | 25.0% | **25.2%** | 27.6% |
| DeepSeek-V3 | 10.1% | **18.0%** | 18.7% | 7.0% | **15.0%** | 15.3% | 7.8% | **14.0%** | 14.8% |
| DeepSeek-V3.1 | 31.1% | **32.7%** | 34.3% | 26.2% | **27.5%** | 29.0% | 23.8% | **25.1%** | 27.6% |
| GPT-4.1 mini | 7.3% | **17.0%** | 17.7% | 5.3% | **11.1%** | 12.0% | 6.0% | **13.1%** | 13.9% |
| GPT-4.1 nano | 9.8% | **14.3%** | 16.7% | 7.1% | **10.0%** | 11.6% | 7.9% | **11.2%** | 13.1% |

Table 4: ConstructiveBench full end-to-end results (Pass@32)

### D.2 CONSTRUCTIVEBENCH AFTER-CUTOFF (106 PROBLEMS)

### D.3 PUTNAMBENCH SUBSET (337 PROBLEMS)

| Conjecturer | DeepSeek-Prover-V2-7B | | | Goedel-Prover-SFT | | | Kimina-Prover-Preview-7B | | |
|---|---|---|---|---|---|---|---|---|---|
| | CoT | ECP | Union | CoT | ECP | Union | CoT | ECP | Union |
| GPT-5 nano | 13.2% | **15.1%** | 18.9% | 11.3% | **12.3%** | 14.2% | 10.4% | **11.3%** | 13.2% |
| GPT-5 mini | 17.9% | **22.6%** | 22.6% | 15.1% | **16.0%** | 17.9% | 13.2% | **14.2%** | 15.1% |
| DeepSeek-V3 | 2.8% | **5.7%** | 5.7% | 1.9% | **4.7%** | 4.7% | 1.9% | **3.8%** | 3.8% |
| DeepSeek-V3.1 | 17.0% | **17.9%** | 20.8% | 15.1% | **16.0%** | 17.9% | 13.2% | **14.2%** | 15.1% |
| GPT-4.1 mini | 4.7% | **6.6%** | 6.6% | 3.8% | **5.7%** | 5.7% | 3.8% | **4.7%** | 4.7% |
| GPT-4.1 nano | 2.8% | **4.7%** | 5.7% | 1.9% | **2.8%** | 3.8% | 1.9% | **2.8%** | 3.8% |

Table 5: ConstructiveBench After-Cutoff end-to-end results (Pass@32)

| Conjecturer | DeepSeek-Prover-V2-7B | | | Goedel-Prover-SFT | | | Kimina-Prover-Preview-7B | | |
|---|---|---|---|---|---|---|---|---|---|
| | CoT | ECP | Union | CoT | ECP | Union | CoT | ECP | Union |
| GPT-5 nano | 1.2% | **1.5%** | 1.8% | 0.9% | **0.9%** | 1.2% | 0.9% | **1.2%** | 1.5% |
| GPT-5 mini | 1.2% | **1.8%** | 1.8% | 0.9% | **0.3%** | 0.9% | 0.9% | **0.6%** | 0.9% |
| DeepSeek-V3 | 0.3% | **0.9%** | 0.9% | 0.0% | **0.3%** | 0.3% | 0.0% | **0.0%** | 0.0% |
| DeepSeek-V3.1 | 1.2% | **1.2%** | 1.5% | 0.6% | **0.3%** | 0.6% | 0.6% | **0.6%** | 0.9% |
| GPT-4.1 mini | 0.6% | **0.6%** | 0.9% | 0.6% | **0.6%** | 0.9% | 0.0% | **0.0%** | 0.0% |
| GPT-4.1 nano | 0.9% | **0.6%** | 0.9% | 0.3% | **0.0%** | 0.3% | 0.0% | **0.0%** | 0.0% |

Table 6: PutnamBench subset end-to-end results (Pass@32) with timeout=120s.

## D.4 FULL ANSWER-CONSTRUCTION ACCURACIES

| Conjecturer | ConstructiveBench (Full) | | ConstructiveBench (After-Cutoff) | | PutnamBench | |
|---|---|---|---|---|---|---|
| | CoT / ECP | Union | CoT / ECP | Union | CoT / ECP | Union |
| GPT-5 nano | 45.1% / **62.3%** | 66.9% | 28.3% / **55.7%** | 57.6% | 16.3% / **32.3%** | 36.8% |
| GPT-5 mini | 69.7% / **73.6%** | 78.9% | 51.9% / **58.5%** | 64.2% | 54.9% / **57.6%** | 71.8% |
| DeepSeek-V3 | 17.3% / **40.7%** | 43.4% | 2.8% / **26.4%** | 26.4% | 19.0% / **31.8%** | 36.8% |
| DeepSeek-V3.1 | 65.4% / **75.0%** | 79.2% | 44.3% / **67.9%** | 69.8% | 53.7% / **62.3%** | 73.6% |
| GPT-4.1 mini | 16.6% / **36.5%** | 37.8% | 11.3% / **23.6%** | 24.5% | 15.1% / **23.2%** | 29.4% |
| GPT-4.1 nano | 15.3% / **28.5%** | 33.0% | 4.7% / **18.9%** | 20.8% | 7.7% / **11.6%** | 17.2% |

Table 7: Answer-construction accuracy. Note that, DeepSeek-V3.1 has knowledge cutoff after June 2024, so this model's ConstructiveBench (After-Cutoff) accuracy may be contaminated.

