# OpenReview forum: "Enumerate-Conjecture-Prove: Formally Solving Answer-Construction Problems in Math Competitions"
_ICLR.cc/2026/Conference — ICLR 2026 Conference Withdrawn Submission_

### Official Review · Reviewer_CAsT · 2025-10-15

**Soundness:** 3
**Presentation:** 3
**Contribution:** 3
**Rating:** 6
**Confidence:** 4

**Summary:**

The paper introduces Enumerate–Conjecture–Prove (ECP), a neuro-symbolic pipeline for answer-construction problems in competition mathematics. ECP (i) enumerates candidate objects via LLM-generated Python, (ii) conjectures a closed-form Lean expression from these hints, and (iii) proves correctness using Lean automation or learned provers (e.g., DeepSeek-Prover-V2-7B). The authors also curate ConstructiveBench, a Lean-verified dataset of 3,640 answer-construction problems gathered and autoformalized with an error-guided loop and an LLM judge. Empirically, ECP improves answer-construction accuracy across models and datasets—for example, GPT-5 mini rises from 69.7% → 73.6% on ConstructiveBench, while end-to-end accuracy increases 32.5% → 33.1%; on the PutnamBench subset ECP solves 6/337 end-to-end vs 4/337 for a CoT baseline.

**Strengths:**

- **Clear problem focus & formulation:** The paper crisply separates theorem-proving from answer-construction and formalizes the latter in Lean’s dependent type theory, including canonical-format constraints to prevent trivial “echo the statement” answers.
- **New dataset with care for quality:** ConstructiveBench (3,640 problems) includes multiple sources, deduplication, a retrieval-aided autoformalization loop, and a post-cutoff test split (106 problems after June 2024) to mitigate contamination. The human study on 100 random items finds 77% fully correct formalizations.
- **Well-structured, modular pipeline:** The three stages—Enumerate → Conjecture → Prove—are easy to reason about and reproduce. Figure 1 illustrates the looped tool-use design; the Prove stage sensibly leverages both Lean tactics and learned provers.

**Weaknesses:**

- **Small end-to-end gains:** While answer-construction accuracy improves notably, end-to-end improvements are modest (e.g., 32.5% → 33.1% on ConstructiveBench; 4 → 6 on PutnamBench). It’s unclear whether these differences are statistically significant or practically meaningful given Pass@32 variance.
- **Compute and sensitivity not fully surfaced:** The enumeration stage caps runtime (60 s) and candidates (≤100), but the paper does not quantify sensitivity to these budgets or how often enumeration misleads conjecturing (the under-generalization can hurt a subset of problems). A cost-vs-accuracy curve and failure taxonomy would help.
- **Limited comparative analysis with concurrent frameworks:** The related work cites broader formal problem-solving frameworks (e.g., FPS/D-FPS) and domain-specific systems, but experiments compare mainly to a CoT baseline rather than to other tool-integrated or search-based pipelines. Including such baselines (where feasible) would clarify ECP’s unique contribution.

**Questions:**

- **Lack of related work:** CounterMath [1] is a counterexample-driven mathematical benchmark that also needs answer construction.
- Line 152: should be a $\neq$ 0.
- Figure 2 is not complete with percentages.



[1] Li, Yinghui, et al. "One Example Shown, Many Concepts Known! Counterexample-Driven Conceptual Reasoning in Mathematical LLMs." Forty-second International Conference on Machine Learning.

---

### Official Review · Reviewer_EgsR · 2025-10-20

**Soundness:** 2
**Presentation:** 3
**Contribution:** 1
**Rating:** 2
**Confidence:** 4

**Summary:**

This paper present the Enumerate-Conjecture-Prove (ECP) framework, a method for solving answer-construction problems in three phases: (1) enumeration of answers satisfying the constraints by hooking up an (informal) LLM to Python, (2) conjecturing of the full solution set using an LLM based on the enumeration, (3) solving the problem in Lean using a formal LLM with the conjecture from step 2 applied. Additionally, the authors present ConstructiveBench: a benchmark consisting of 3640 formal answer-construction problems taken from public high-quality competitions. The authors show in their experiments that ECP outperforms a baseline that just uses steps 2 and 3.

**Strengths:**

- Code is provided, well-documented, and well-structured.
- Paper is clean and easy to read and understand.
- The authors investigate an important area that is lacking in current formal reasoning, namely coming up with the answer.
- The authors slightly outperform a reasonable baseline.

**Weaknesses:**

My main concerns with this paper is the lack of any interesting contribution. I have structured each of my concerns in three categories: major weaknesses, weaknesses, and remarks. The latter have not influenced my judgment, but should be fixed.

**Major Weaknesses**
- **ConstructiveBench is very limited**, and just another formal benchmark with little effort taken in its creation (a simple, cheap pipeline is used that the authors themselves show contains significant noise). As such, it provides no benefit over already existing benchmarks such as MiniF2F and PutnamBench. In fact, it is worse than these benchmarks because of three reasons:
   - Noise: 23% noise is high, and no measure is taken to reduce this. PutnamBench has very little noise since its constructed by human experts.
   - Contamination: By getting problems from existing public competitions, significant contamination concerns pop up. The authors even admit so by using their "past june 2024 selection". While PutnamBench also has this problem, it has several advantages that make this less concerning: it has been a well-known benchmark for years, leading to decontamination procedures for existing models are likely implemented, it has been selected from a single competition, making exclusion of these problems in training much easier, it is constructed by human experts, and does not use the same autoformalizers that are also used by the LLMs for their training.
   - While ConstructiveBench likely has more answer construction problems, the authors show that a significant number can also be found in PutnamBench.
- **ECP is simple and leads to very low improvements.** The only novel thing in ECP is the "E" stage (as the authors themselves admit by including "CP" as a baseline). However, enumerating answers is very limited and applicable to only a limited number of problems where such enumeration can be performed. While it can be made more general (e.g., for proving an upper bound, a model could try various parameters and see what the maximum is it gets), the system prompt used by the authors make it clear they did not envision this: "Your task is to write a Python program to enumerate possible answers to assist in conjecturing and proving the true answer." Furthermore, the E stage can be very easily put into the C stage by just allowing tool use by the model. In fact, the implementation, even in the E stage, should use such a tool-calling implementation, especially for reasoning models. Currently, the authors parse the python in the result of the model output manually, which is suboptimal. Finally, the numbers only increase by only up to 2% (only looking at reasoning model results, the others I am discarding as they are not relevant, see weaknesses). This is very limited even if the authors proposed a very nice and novel method.

**Weaknesses**
- **Informal models are given very limited reasoning tokens.** This is especially problematic given that high reasoning effort would likely result in models performing the enumeration stage more manually as well, with specific directions being explored. 4000 tokens is not enough for this. Additionally, the authors should include at least one state-of-the-art model (grok 4, gemini 2.5 pro, gpt-5), not just mini variants. Finally, non-reasoning models are unnecessary: they are essentially never relevant for any mathematical task.
- **Compute limit should be equal between baseline and ECP**. Since the first stage is skipped for the baseline, it essentially uses a lower amount of compute than ECP. I am not entirely sure how to most appropriately fix this (and whether it even can), but at the very least, the number of iterations allowed for it should be 8 instead of 5. Note that this comment is very much on the limit between remark and weakness, as I do not think it will influence much.
- **The differentiation between answer-construction and theorem-proving tasks is somewhat artificial in places**. Many theorem proving tasks also require you to come up with answers. I do understand where the authors come from: in formal theorem proving, the answer for such theorem proving tasks is always already given in the theorem statement that needs to be proven. However, the way it is presented now makes it seem as if this is also often the case in natural language mathematical problems, which is most definitely not the case. For instance, the comment in L40-44 is somewhat out of place for the motivation here: most of the USAMO problems also require finding an answer and then proving it. The authors should more clearly differentiate this. Personally, I think it is much easier to argue the necessity by simply noting that formal theorem proving almost always skips the "finding the answer" stage of a mathematical problem.
- **The use of the term "answer-construction" problems is somewhat strange here and there**. Final-answer problems is a more agreed upon term in the literature, and answer-construction makes it seem as if the authors are only handling constructive problems. Interestingly, they also define answer-construction problems as constructive problems in L138-139. However, the more important issue is that solving any final-answer problem is skipped in formal reasoning, not just for those where enumeration can take place. For instance, finding the maximum value of a certain expression, the number of ways in which something can be done, ... all fall under final-answer problems, but not necessarily under problems where enumeration is possible (for constructive problems). I believe the authors do mean the "final-answer" interpretation of "answer-construction" problems, as becomes clear in L155-170, but in this case the use of an enumeration stage becomes somewhat problematic as it is not relevant for many problems.
- The authors should use the recommended hyperparameters for all models. For instance, Deepseek-v3.1 is advised to be run with temperature 0.6, not temperature 1.0.

**Remarks**
- L152 and L154 in agent.py in the code reads two files that do not exist, should be conjecturer_formal.txt?
- L463-465 is not true: DeepMind got silver with a formalized approach last year, SeedProver got gold this year (although they got P1 only after the grading was finished).

**Questions:**

See above

---

### Official Review · Reviewer_7WCo · 2025-10-25

**Soundness:** 2
**Presentation:** 2
**Contribution:** 2
**Rating:** 4
**Confidence:** 3

**Summary:**

This paper is based on the observation that LLMs, when guided by a careful prompting and tool integration, excel in solving high-school mathematics problems, but they struggle in mathematical reasoning problems requiring formal proof or the construction and verification of complex mathematical objects. For mathematical reasoning, LLMs can tackle difficult answer-construction tasks but are prone to errors from hallucinations and unverifiable steps. On the other hand, symbolic methods guarantee rigor but falter in creative answer construction. Motivated by these observation, this paper aims to address the following gap: “how to solve answer- construction problems while preserving both LLM creativity and mathematical rigor?”

Towards bridging this gap, this paper proposes Enumerate–Conjecture–Prove (ECP) framework, a modular neuro-symbolic method. ECP is model-agnostic and shows consistent improvements over pure LLM baselines. ECP framework is more like an agentic flow where an LLM is invoked for enumerating candidate answers. The candidate answer enumeration is achieved by means of LLM generating a python program.
This paper seems to be focusing on two kinds of problems in math competitions: answer-construction problem and theorem-proving problem. For the answer-construction problems, it proposes a mapping of the task into the Lean dependent type framework.

Next, this paper also proposes a dataset called ConstructiveBench. This is a collection of answer-construction problems and is sourced from multiple original sources including AMC 12 A/B  (from 2000 onward), AIME I/II, HMMT, regional Olympiads, and the IMO Shortlist and Longlist.  Additionally, this dataset integrates data from established informal math datasets including OlympiadBench (He et al., 2024), Omni-Math (Gao et al., 2024a), and MathOdyssey (Fang et al., 2024). This paper also proposes an autoformalization pipeline translate each informal answer-construction problem into a formal version.

To show the efficacy of ECP framework, this paper evaluates it on the proposed ConstructiveBench and PutnamBench datasets; and compares it against classic chain-of-thought (CoT) baseline method. On a subset of PutnamBench for answer construction, ECP formally solves 6 out of 337 answer-construction problems end-to-end (up from 4 without ECP) using GPT-5 mini and DeepSeek-Prover-V2-7B. On the ConstructiveBench, ECP achieves 33.1% end-to-end state-of-the-art accuracy (up from 32.5%).

**Strengths:**

- The constructiveBench dataset could be useful to the community for advancing neuro-symbolic systems.
- ECP improved the state of the art on ConstructiveBench, raising GPT-5 mini’s answer-construction accuracy from 69.7% to 73.6% and its end-to-end problem-solving accuracy from 32.5% to 33.1%.

**Weaknesses:**

- It was unclear whether the focus of the paper is only on answer-construction problems or also on theorem-proving problems. Most of the discussions seem to be focusing only on answer-construction problem. Its better to clarify this.
- Section 3.2 is bit cryptic and somewhat less clear. There is a scope of improving the writing to make it more accessible.
- Figure 3 is not referred anywhere in the text. I, as a reader, found myself lost when reading Section 4.3 because it started in a bit abrupt manner and there was no reference to the example given in Figure 3 for motivating the section. Some rewriting would help improving this section.

**Questions:**

- In Line 152, should it be $a \ne 0$ instead of $b \ne 0$?
- From the given illustration of ECP in Section 3.2,  it is not quite clear what is the difference between Enumerate and Conjecture stages. Looks to me that Enumerate stage itself is sufficient for the answer-construction kind of tasks?

---

### Official Review · Reviewer_1TmT · 2025-10-28

**Soundness:** 2
**Presentation:** 2
**Contribution:** 2
**Rating:** 2
**Confidence:** 4

**Summary:**

This paper addresses the challenge of solving "answer-construction" mathematical problems, which require both the creative generation of a mathematical object and a rigorous proof of its correctness. The authors introduce the `Enumerate-Conjecture-Prove (ECP)` framework, a modular neuro-symbolic method. `ECP` first uses an LLM to generate and execute Python code to enumerate potential solutions, then uses these examples to guide the conjecture of a closed-form answer in Lean, and finally attempts to formally prove the result using a Lean prover. To support this work, the paper also presents `ConstructiveBench`, a new dataset of 3,640 answer-construction problems autoformalized into Lean via an iterative, retrieval-augmented process. The authors report that this process achieves a 77% correctness rate in a human audit. Experiments show that `ECP` yields consistent, though modest, gains over a CoT baseline, improving the end-to-end solve rate on `ConstructiveBench` from 32.5% to 33.1% and increasing the number of solved `PutnamBench` problems from 4 to 6.

**Strengths:**

1. The effort for arriving at a correct autoformalization is commendable, involving multiple autoformalization models, retrieval of relevant documentation and Lean entitites, and a feedback loop.

 2. The presented formalized dataset has significantly lower error rate than those presented in prior work. Further, including a human validation audit and an "after-cutoff" subset to address data contamination concerns adds significant credibility to the benchmark.

 3. The `ConstructiveBench` dataset is a significant new resource, notable for its scale (3,640 problems) and its focus on answer-construction tasks.

 4. The modular nature of the `ECP` pipeline allows for incorporating future advances in models and provers.

**Weaknesses:**

1. The primary weakness is that the empirical improvements from the complex `ECP` pipeline are marginal, especially for state-of-the-art models (e.g., a 0.6% absolute gain on `ConstructiveBench`). The paper provides no confidence intervals or statistical significance tests, making it difficult to determine if these small gains are meaningful or simply due to noise.

2. The different components of the pipeline are not separately evaluated. It is crucial that the authors perform ablation studies to verify how the removal/addition of each component in the `ECP` and `ConstructiveBench` pipelines affect both the dataset and results.

3. Crucial details are missing, hindering the paper's clarity and reproducibility. Specifically:
    - The mechanism for verifying against vacuous or trivial answers in the "Conjecture" stage is not explained.
    - The method for formally checking the equivalence of a conjectured answer and the ground truth is underspecified.
    - The exact format for how Lean compiler messages and retrieved definitions are passed back to the model during iterative refinement is not provided.

4. While the creation effort is a strength, the result has limitations. The human audit revealed a 17% "major error" rate in the final dataset. This is a non-trivial error rate for a "verified" benchmark and raises concerns about the validity of the evaluation results, as models are being tested on a significant number of incorrectly formalized problems. The reliance on an LLM judge for the final semantic check also risks introducing systematic, hard-to-detect biases.

5. The related work section rarely relates to the proposed paper, making it difficult to contextualize the work in the scope of the broader field. Further, the authors might find it beneficial to emphasize why the problem they are working on is important and namely how the answer-construction tasks differ from theorem-proving ones (given that in most benchmarks the former can be rephrased into the latter).

**Questions:**

1. What was the nature of the "major" and "minor" errors found during the human validation of `ConstructiveBench`? For instance, did errors typically make statements more permissive, factually incorrect, or unprovable? What were the experimental outcomes (for both CoT and `ECP`) on these incorrectly formalized problems?

2. It is unclear how much each component in the autoformalization pipeline affects the quality of the formalized statements. Can the authors provide an ablation study or evidence from prior work to quantify how much each component of the autoformalization pipeline (retrieval, feedback, LLM judge) contributes to the final formalization quality?

3. Can the authors present 95% confidence intervals on their results, so that readers are informed about the statistical significance of the results?

4. What are the precise differences between the CoT and `ECP` frameworks in the experiments? The paper implies CoT also involves conjecturing and proving. Is the only difference the "Enumerate" stage? How does `ECP` compare to a simpler baseline that only uses a prover on a ground-truth formalization with the answer provided?

5. How is equivalence between the conjectured and ground-truth answers verified programmatically?

6. How is the check against vacuous answers (L214-215) implemented?

7. Could the authors include the full prompts for the LLM judge, the Lean solver/prover, and the iterative refinement loop in the Appendix?

8. How does this work compare methodologically and empirically to recent work like Hilbert [1], which also combines informal reasoning with formal methods?

9. What are the baseline results on the PutnamBench problems when the final answer is provided (as in the original benchmark)? This is crucial for judging the value of an end-to-end pipeline that must discover the answer itself.

10. The autoformalizations use Lean v4.23.0, while many prover models were trained on earlier versions (e.g., v4.15.0). Have the authors verified that this version mismatch does not negatively impact prover performance?


## Current recommendation

I have assigned this paper a score of **2: Reject**. I believe the paper's core contributions are potentially valuable to the field. However, the paper in its current form suffers from significant weaknesses that undermine its claims. The empirical gains are marginal without statistical validation, critical implementation details are omitted, and there is no ablation to justify the framework's complexity or comparison to key alternative methods. If the authors can thoroughly address my concerns, I would be happy to raise my score.

### References

[1] Varambally, Sumanth, et al. "Hilbert: Recursively Building Formal Proofs with Informal Reasoning." arXiv preprint arXiv:2509.22819 (2025).

---

### Note · Authors · 2025-11-23

I have read and agree with the venue's withdrawal policy on behalf of myself and my co-authors.